# Mental Imagery in the Relationship between Alexithymia and Parental Psychological Control

**DOI:** 10.3390/bs14030183

**Published:** 2024-02-26

**Authors:** Zengjian Wang, Ziying Yang

**Affiliations:** 1Key Laboratory of Brain, Cognition and Education Sciences, South China Normal University, Ministry of Education, Guangzhou 510631, China; 2Guangdong Key Laboratory of Mental Health and Cognitive Science, School of Psychology, Center for Studies of Psychological Application, South China Normal University, Guangzhou 510631, China

**Keywords:** parental psychological control, visual mental imagery, alexithymia, mediation analysis

## Abstract

This study aims to explore the mediating role of mental imagery in the relationship between alexithymia and parental psychological control among Chinese university students. Conducted between March and April 2023, this descriptive study involved 282 volunteer participants from a university in southern China. Data collection included the Toronto Alexithymia Scale (TAS), the Parental Psychological Control Scale (PPC), and the Vividness of Visual Mental Imagery questionnaire (VVIQ). The results revealed that: (1) based on established cut-off, 81 students were identified as highly alexithymic; (2) the alexithymia group scored higher on both the TAS and PPC and lower on the VVIQ compared to the non-alexithymia and possible-alexithymia groups; and (3) mediating analysis demonstrated a strong and positive correlation between parental psychological control and alexithymia for all participants, with visual mental imagery mediating this relationship. This study underscores the interconnectedness of parental psychological control, visual mental imagery, and alexithymia among college students. The theoretical and clinical implications of these findings are also discussed.

## 1. Introduction

Alexithymia, originally defined as “without words for feelings”, is a multifaceted personality construct that refers to one’s inability to successfully deal with emotional regulation [1]. This trait encompasses difficulties in identifying, describing, and articulating one’s emotions, a limited capacity for imaginative thinking, and a tendency towards externally oriented thought processes [2,3]. Individuals with alexithymia are aware that they are experiencing an emotion but often find it challenging to pinpoint the specific emotion they are going through [4]. Researches have indicated that alexithymia is associated with a range of conditions, such as eating disorders, depression, schizophrenia, autism, and substance abuse, and it is also commonly observed in the general population [5]. Studies of alexithymia using a cut-off point from the Toronto Alexithymia Scale (TAS-20) have reported rates ranging from 15.7% to 31.9% in Chinese college students [6,7,8,9]. Given that different values are attached to emotional awareness and expression in different countries, there might be some variations across different cultures [10,11,12,13,14]. Many factors including family environment and cognitive capacity have been shown to be important for individuals’ emotional reactivity and emotional regulation [6,9,15,16,17,18,19]. Previous studies have demonstrated that parenting styles, particularly controlling styles, are positively correlated with alexithymia [14,20]. However, there is limited research investigating the underlying mechanisms of how parenting control influences alexithymia through cognitive function [21].

### 1.1. Parental Psychological Control and Alexithymia

Psychological control refers to a parenting practice marked by intrusive and manipulative behaviors intended to shape children’s or adolescents’ thoughts and emotions, allowing adults to exert control over their psychological world. Such behaviors encompass guilt induction, withholding affection, and manipulating the parent–child relationship [22]. Employing parental psychological control creates an environment that restricts the capacity of youths to cultivate self-awareness and comprehension of their own emotions [21]. Numerous studies have delved into the direct link between parental psychological control and alexithymia [14,23].

According to the self-determination theory [24], parental psychological control has been linked to the development of alexithymia, often exacerbated by elevated instances of child maltreatment [25] and parental overprotection [26], and anxious attachment [20,27], as well as a reduction in psychological needs and the disruption of self-regulation [28]. Moreover, parents with alexithymic traits were found to show more parental psychological control, and may encounter challenges in comprehending their children’s emotional experiences [14]. Children’s perception of a neglectful parenting style affects their emotional awareness [29]. Moreover, in accordance with the social–cognitive model, a previous study [30] has provided evidence that emotional schemas function as a partial mediator in the link between perceived parental styles and alexithymia in adult Iranian migraine patients. A more recent study using latent class analysis found that individual and familial factors interact to affect adolescents’ mental health [28]. Given that parental psychological control is often associated with emotional maladjustment, it may be particularly important to understand the mechanism of parental psychological control in youths’ alexithymia [31].

### 1.2. Mental Imagery and Human Emotion

Mental imagery refers to representations and the accompanying experience of sensory information without a direct external stimulus [32]. Mental images play an important role in emotional processes and are closely involved in emotional regulation [33]. It has been proposed as an “emotional amplifier” of mental content, and individuals with limited mental imagery capacity tend to display reduced emotional responsiveness [34]. Moreover, individuals grappling with emotional disorders often manifest irregularities in their mental imagery, including an excess of distressing intrusive negative imagery, a deficiency of positive imagery, a bias toward observer perspective imagery, and a lack of specific imagery as observed in overgeneral memory [35].

The vividness of mental imagery refers to the clarity, brightness, or intensity of an image as reported by the individual [36]. Previous studies have shown that the higher levels of vividness in negative mental imagery are associated with greater distress [35,37]. Intensely vivid mental imagery is prevalent across a spectrum of psychopathologies, encompassing anxiety disorders [38], post-traumatic stress disorder [39], schizophrenia [32], and substance use disorder [40]. Alexithymia has also been associated with challenges in affect-related mentalization [41]. Earlier research has demonstrated that individuals with alexithymia exhibit lower mental imagery capacity as measured by self-reported vividness of visual mental imagery when compared to those without alexithymia [42]. Mantani et al. [43] reported reduced activation of the posterior cingulate cortex during future happy imagery in subjects with high degrees of alexithymia. Gay and colleagues [44] discovered that the hypnotic imagery intervention proved to be an effective method for directly reducing alexithymic scores, without specifically addressing a decrease in anxiety or depression scores. However, there was a study showing no significant differences between high and low alexithymic women in script-driven emotional imagination ability, electrodermal activity during imagination, and the ratings of valence, arousal, and vividness [45]. A more recent study [46] highlights the predictive capacity of alexithymia concerning deficits in processing arousal-based emotions and the presence of discordance between emotional response systems during emotional imagery. Their findings suggest that while alexithymia demonstrated limited associations with valence-based measures such as facial electromyography and valence ratings, it exhibited significant connections with arousal-based measures. Accordingly, more studies are needed to reveal the relationship between the vividness attribute of mental imagery and alexithymia.

As we explore the relationship between parental psychological control and mental imagery, we find a limited number of studies directly investigating how the former impacts the latter. In a recent study by Kim, Baik, and Nam [47], it was revealed that adolescents’ self-perceived parenting attitudes significantly influence their cognitive performance. This influence is notably reflected in a decline in visuo-spatial attention ability and the adoption of an impatient strategy for mental imagery. The study further proposes that the negative evaluation of parenting attitudes by adolescents may trigger a discernible decrease in cognitive abilities, such as visuo-spatial attention and mental rotation, before the formal diagnosis of associated mental disorders occurs. Among adolescents diagnosed with post-traumatic stress disorder who had experienced child abuse, including physical abuse by a parent or caregiver, reported significantly higher frequencies of negative mental imagery. These images were characterized as being more vivid, distressing, and tightly intertwined with their autobiographical memories [48]. However, the vividness of parental mental imagery was positively associated with both parental acceptance of the child and autonomy support, along with various aspects of the climate for creativity in the parent–child relationship [49].

Expanding upon the existing body of research that underscores the connection between parental psychological control, mental imagery, and alexithymia, it is imperative to undertake an empirical investigation into their underlying mechanisms. Consequently, the primary objective of this study is to examine whether college students exhibiting high levels of alexithymia also demonstrate elevated levels of parental psychological control and reduced vividness in visual mental imagery. Additionally, this study puts forth the following hypotheses: that the vividness of mental imagery serves as a mediating factor in the influence of parental psychological control on alexithymia.

## 2. Materials and Methods

### 2.1. Procedure

This study obtained approval from the local Ethics Committee of the School of Psychology, South China Normal University (SCNU-PSY-2021-020). Participants were recruited through social media platforms, primarily WeChat. An online survey was designed using Wenjuanxing (https://www.wjx.cn) and shared on social media. The online survey was initiated with an informed consent process, during which students were informed about the survey’s focus on exploring the impact of emotional difficulties. Subsequently, the participants responded to demographic questions, including gender (male or female), grade level (freshman, sophomore, junior, senior), and whether the participant is an only child or not, followed by self-reported measurements. Data collection occurred from March 2023 to May 2023. Convenient sampling was employed to recruit participants from universities, and no alternative recruitment strategies were utilized. University students from four years of undergraduate study were invited to participate in this study. Students who refused to participate and forms with the wrong answer to the question “somewhat agree” were excluded.

### 2.2. Sample Size Calculation

Sample size was calculated using G*Power version 3.1.9.7. Due to the different statistical methods applied, we set the parameters for the test with the highest sample size requirement, that is, an F test. Setting a medium effect size (f  =  0.25), an α-error probability of 0.05, a power of 0.95, and 3 as the number of groups, the required sample size was 252 participants. A total of 336 Chinese college students participated in the study. Seventeen participants were excluded from the analysis because they chose incorrect answers when instructed to select ‘somewhat agree’. Outliers (*n* = 23) were defined as those for whom the discrepancy between self-reported parental psychological control, vividness of visual mental imagery, and alexithymia exceeded two standard deviations from the mean discrepancy. The identification and removal of outliers served to remove their potentially undue influence during correction generation. The results were the same with and without outliers. The final analysis included 282 participants (52.50% female). The participant distribution was mainly composed of freshman (0.7%), sophomore (6.4%), junior (15.2%), and senior students (77.7%), with 52.8% being the sole child in their families.

### 2.3. Research Instruments

#### Parental Psychological Control (PPC)

The evaluation of parental psychological control utilized the 18-item Parental Psychological Control Scale [50]. This scale measures three key dimensions of parental psychological control: authority assertion, guilt induction, and love withdrawal. Authority assertion is assessed through three items, such as “My parents emphasize that I should not argue with them”. Guilt induction is evaluated using ten items, including statements like “My parents tell me that I am not a good member of the family without meeting their expectations”. Love withdrawal is measured using five items, such as “My parents are less friendly with me if I do not see things their way”. Respondents were instructed to rate the frequency of their parents’ behaviors related to these dimensions using a 5-point rating scale, with 1 indicating ‘never’ and 5 indicating ‘always’. Individual item scores were averaged to create a composite index, with higher scores indicating a stronger perception of parental psychological control. It was found that the scale is suitable for college students for self-assessment, with good reliability and validity, and the scale has been revised and tested many times [51,52,53]. In our study, the internal consistency of this assessment tool was found to be 0.98.

### 2.4. Vividness of Visual Imagery (VVIQ)

Each participant in our study completed the Vividness of Visual Imagery Questionnaire [54,55], a well-established tool widely used in the field [56,57,58]. The VVIQ consists of 16 items, requiring participants to assess the vividness of their mental imagery related to various scenarios, including familiar individuals, shops, the sky, and countryside scenes (e.g., imagining a sunrise transitioning into a rainstorm). Participants provided ratings on a scale ranging from 1 (no image at all, just thoughts about the object) to 5 (extremely clear imagery, as if the object were actually in sight). Scores on this rating system can vary between 16 and 80. Typically, research employing the VVIQ as a measure of visual imagery asks participants to complete the questionnaire twice, once with their eyes open and once with their eyes closed. The mean average scores under both conditions were used to calculate the VVIQ index, with a higher total score on the VVIQ indicating a greater ability to generate vivid mental imagery. In our study, the VVIQ demonstrated good reliability, with a Cronbach’s α of 0.83 for the condition with open eyes and 0.79 for the condition with closed eyes.

### 2.5. Toronto Alexithymia Scale (TAS-20)

The Chinese version of the 20-item Toronto Alexithymia Scale (TAS-20) [59] is a self-report scale for the assessment of alexithymia. It comprises 20 items that are categorized into three factors: (1) difficulty identifying feeling (DIF, 7 items; e.g., “I have feelings that I can’t quite identify”); (2) difficulty describing feelings (DDF, 5 items, e.g., “It is difficult for me to reveal my innermost feelings, even to close friends”); (3) externally oriented thinking (EOT, 8 items; e.g., “I prefer to watch “light” entertainment shows rather than psychological dramas”). Respondents rate each item on a 5-point Likert scale, ranging from 1 (strongly disagree) to 5 (strongly agree). Notably, items 4, 5, 10, 18, and 19 are scored in reverse. The total TAS-20 score falls within a range of 20 to 100, with higher scores indicating more severe alexithymia. A score of 51 or lower suggests the absence of alexithymia, scores between 52 and 60 indicate potential alexithymia, and scores of 61 or higher suggest the presence of alexithymia [60,61,62]. Previous research has demonstrated that the Chinese version of the TAS-20 had good reliability and validity in undergraduates [63]. In the current research, the Cronbach’s alpha value was 0.87.

### 2.6. Data Analysis

SPSS 22.0 software was used in the analysis. Tests of normality revealed that the study variables showed no significant deviation from normality (i.e., skewness < |3.0| and kurtosis < |10.0| [64]. To address the potential common method bias associated with self-administered questionnaires, this study utilized Harman’s single-factor test [65] and performed an exploratory factor analysis on all of the items that covered the variables.

We first carried out descriptive statistics on demographic variables and three variables and then standardized the data of the three variables. One-way ANOVA was used to test the differences in PPC, the VVIQ, and the TAS among the three groups. To test the hypothesis, we used Pearson correlation analysis to explore the relationship between parental psychological control, vividness of imagery, and alexithymia. Then, we used PROCESS macro 4.1 software [66] to examine the mediating role of vividness of imagery between parental psychological control and alexithymia, which was specifically developed for testing the complex models. We followed Baron and Kenny’s recommendations regarding the design of a mediation model, according to which the independent variable is required to predict the mediator and the mediating variable is required to predict the dependent variable. We used a bias corrected bootstrap technique with 5000 samples and a set of 95% confidence interval. Previous studies have demonstrated that gender, grade, and only-child status have an impact on the levels of alexithymia [10,67,68]. Consequently, gender, grade, and only-child status were included as covariates in the model.

## 3. Results

### 3.1. Common Method Bias Test

Harman’s single-factor test showed that the first factor accounted for 25.58% of the total variation, lower than the threshold of 40% proposed by Podsakoff et al. [69]. Although this result does not eliminate the possibility of common method variance, it suggests that common method bias is unlikely to confuse the interpretation of the data analysis results.

### 3.2. Descriptive Analysis and Correlations among Overall Variables

Table 1 presents the means, standard deviations (SDs), and differences among the three groups. The one-way ANOVA conducted on the three groups revealed statistically significant differences in relation to the three variables (Table 1). Subsequent post-hoc analyses clarified that individuals in the alexithymia group exhibited significantly higher levels of TAS compared to both the possible-alexithymia group and the non-alexithymia group (*p*s < 0.01). Furthermore, the possible-alexithymia group also demonstrated higher levels of TAS compared to the non-alexithymia group (*p* < 0.001). Additionally, the alexithymia group displayed the lowest VVIQ scores, significantly differing from both the possible-alexithymia group and the non-alexithymia group (*p*s < 0.01). The non-alexithymia group scored higher on the VVIQ compared to the possible-alexithymia group (*p* = 0.003). Regarding PPC, the alexithymia group exhibited the highest levels in comparison to the other two groups (*p*s < 0.001), while the possible-alexithymia group displayed higher PPC levels than the non-alexithymia group (*p* = 0.006).

Pearson r correlations among all variables are presented in Table 2. Across all participants, significant correlations were noted between the VVIQ and both PPC and TAS. Noteworthy correlations were identified between the VVIQ and TAS in both the alexithymia and non-alexithymia groups. No significant correlations were detected between the VVIQ and the remaining variables within the possible-alexithymia group.

### 3.3. The Mediating Effects Analyses

The college students’ VVIQ answers were then examined as a mediating variable between overall TAS and PPC (Figure 1). Indirect effect analysis showed that PPC has a significant overall effect on TAS [(c: B = 0.39, SE = 0.05, CI (0.29, 0.49)]. With the effect of the VVIQ on TAS as a mediator [a: B = −0.24, SE = 0.06, CI (−0.35, −0.13); b:B = −0.35, SE = 0.05, CI (−0.44, −0.25)], the direct effect of PPC on TAS decreased [(c’: B = 0.31, SE = 0.05, CI (0.21, 0.40)]. The indirect effect of PPC on TAS through the VVIQ [(B = 0.08, SE = 0.02, CI (0.04, 0.13)] denotes partial indirect effect by the mediator (VVIQ).

## 4. Discussion

The primary aim of this research was to investigate the mediating role of visual mental imagery in the connection between parental psychological control and alexithymia among Chinese college students. In general, the proposed model was found to be suitable for the data and indicated that mental imagery partially mediates the mentioned relationship.

In our sample of Chinese college students, we observed a significant prevalence of alexithymia, reaching 28.72%. This prevalence aligns closely with findings from previous studies conducted in diverse cultural contexts, such as China (31.9% among nursing students) [6], Italy (27.3% among college students) [70], Jordan (24.6% among university students) [71], New York (27.8% for men and 26.8% for women among college-aged students) [72], and Egypt (22% among university students) [73]. Indeed, an increasing number of studies have documented a heightened prevalence of alexithymia in the general population [74] following the COVID-19 pandemic, surpassing the pre-pandemic rate of around 10% [75,76]. Moreover, this trend extends to mental health disorders including depression [77,78]. While it may not align consistently with the notion of whether alexithymia is a stable or unstable personality trait, it is evident that external pressures can indeed elevate the levels of alexithymia [79,80,81]. Longitudinal studies are essential to explore the development of alexithymia among diverse cohorts, providing a comprehensive understanding of approaches to managing the health challenges faced by individuals with pronounced alexithymic personality traits [82,83].

In line with previous studies [21,23,28], our findings also reveal a positive relationship between parental psychological control and alexithymia. This suggests that parenting styles may impact how children learn to label, identify, and regulate emotions through the process of emotion socialization. Furthermore, we observed a negative relationship between the vividness of visual mental imagery and the levels of alexithymia, specifically in the difficulty in identifying feelings within the high alexithymia group. Our results align with a previous study that discovered lower vividness of visual mental imagery in individuals with high levels of alexithymia [42]. Samur et al. [84] also identified a notable negative association between the mental imagery subscale of the Story World Absorption Scale and alexithymia. A plausible explanation is that the compromised construction of internal representations of emotional stimuli in alexithymia may serve as a mechanism explaining their challenges in emotional processing [85]. This notion aligns with the observation that visual thoughts tend to evoke more intense emotions than verbal thoughts [86], and the utilization of imagery has been demonstrated to amplify emotional physiological responses [34]. Nevertheless, since the measurement of mental imagery ability relied on self-reporting, further studies utilizing objective perceptual measures of imagery, such as the binocular rivalry technique [87], are necessary to investigate the relationship with alexithymia.

Additionally, our results suggest that visual mental imagery could act as a partial mediator in the link between parental psychological control and alexithymia. To the best of our knowledge, our study is the first to investigate the mediation mechanism underlying parental psychological control and alexithymia among healthy Chinese college students. Prior to our research, only one study delved into the underlying mechanism of parental styles and alexithymia. Hormozi and colleagues [30] discovered that emotional schemas partially mediate the relationship between parental styles and alexithymia in migraine patients. The relevance of mental imagery to psychopathology is acknowledged, given its presumed unique connection with emotion [86,88]. There is a need for increased efforts to incorporate imagery into therapeutic interventions aimed at modifying negative emotional conditions.

Our work has several limitations that can be addressed in future work. Firstly, the investigation did not assess other variables linked to psychiatric disorders, including anxiety or depression, potentially introducing confounding bias. Subsequent studies should explore whether other mental health conditions influence the relationship between mental imagery and alexithymia. Secondly, the measurement of the vividness of visual mental imagery relied on self-reports, and the evaluation of controllable mental imagery in individuals with alexithymia was relatively straightforward. This method may not fully capture the intricate nature and significance of mental imagery vividness, such as the uncontrollable and intrusive imagery experienced by individuals with alexithymia. To bolster the robustness of the findings, future studies are encouraged to employ diverse measurement approaches. Thirdly, this research is constrained by objective factors and employs a cross-sectional study design. While previous research forms a strong foundation for this study, it does not completely elucidate the mechanisms through which parental psychological control and mental imagery influence alexithymia. In future investigations, a longitudinal research design could be employed to explore the causal relationships between these variables.

In conclusion, by addressing these limitations and incorporating more robust research methods, future studies can offer a more comprehensive understanding of how parental psychological control and mental imagery impact the alexithymia of Chinese college students. This, in turn, could significantly contribute to the formulation of more effective strategies for emotional regulation within this population.

## Figures and Tables

**Figure 1 behavsci-14-00183-f001:**
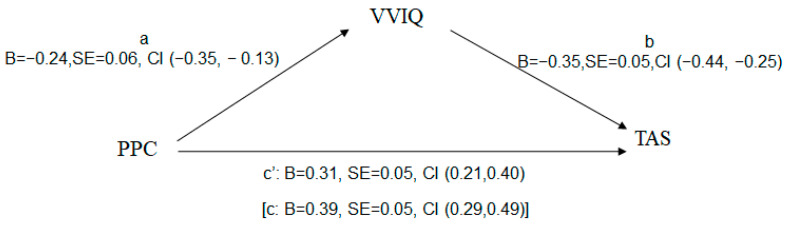
Mediating effect of visual mental imagery between parental psychological control and alexithymia. VVIQ, vividness of visual mental imagery; PPC, parental psychological control; TAS, alexithymia.

**Table 1 behavsci-14-00183-t001:** Difference in the variables among the three groups.

	Group			F
	Non-Alexithymia (*n* = 103)	Possible-Alexithymia (*n* = 98)	Alexithymia (*n* = 81)	
TAS	43.72 ± 5.90	57.16 ± 2.20	66.67 ± 4.01	647.03 **
PPC	51.62 ± 18.14	59.21 ± 21.49	73.75 ± 18.70	29.52 **
VVIQ	55.46 ± 6.60	52.76 ± 6.83	48.64 ± 6.50	26.83 **

Note: **, *p* < 0.01; TAS, Toronto Alexithymia Scale; PPC, Parental Psychological Control; VVIQ, Vividness of Visual Imagery Questionnaire.

**Table 2 behavsci-14-00183-t002:** Pearson *r* correlations among the study variables.

	All Participants (*n* = 282)		Alexithymia Group (*n* = 81)		Possible-Alexithymia Group (*n* = 98)		Non-Alexithymia Group (*n* = 103)	
	VVIQ	PPC	VVIQ	PPC	VVIQ	PPC	VVIQ	PPC
TAS	−0.44 **	0.40 **	−0.28 **	0.27 **	−0.14	0.04	−0.21 *	0.02
PPC	−0.27 **	-	−0.39 **	-	−0.10	-	0.06	-

Note: **, *p* < 0.01; *, *p* < 0.05; TAS, Toronto Alexithymia Scale; PPC, Parental Psychological Control; VVIQ, Vividness of Visual Imagery Questionnaire.

## Data Availability

The data presented in this study are available on request from the corresponding author.

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
