# Peer review of "Mental Imagery in the Relationship between Alexithymia and Parental Psychological Control"

_behavsci, 2024, doi:10.3390/bs14030183_

Round 1
Reviewer 1 Report
Comments and Suggestions for Authors
This manuscript describes a study that brings together three important factors - parental control, mental imagery vividness and alexythymia. It is generally well written and a would make a good scientific contribution. However, it has a flaw in the presentation of the analyses, and it is missing sufficient considerating of a conceptual/psychometric issue.
The issue with the analysis is that the tables and text do not show the steps that one would expect from a mediation analysis, and conclusions are made within the text of the Results that intepret and attempt to enhance the meaning of the findings in an overgeneral manner.
This sentence is particularly telling: ". When introducing VVIQ into the regression equation, the predictive relationship between PPC and 274 TAS remained statistically significant (β = 0.33, p < 0.01). This underscores the intricate interplay between PPC, VVIQ, and TAS, thus further supporting the notion that visual mental imagery potentially mediates the relationship between parental psychological control and alexithymia." This is both statistically incorrect, and not appropriate for a Results section. It is not the case that by adding an additional predictor, the maintenance of an existing effect illustrates that the additional predictor is a mediator. I also cannot work out how the following table illustrates mediation. I think it would be easier for the authors to follow the simple Baron and Kenny approach and spell out the steps in the text.
The conceptual/psychometric issue is that vivid controllable mental imagery is different from vivid intrusive and uncontrollable mental imagery. I accept that the VVIQ may not measure this, but arguable a new measure needs to be identified. It is often the case that the very same individuals who struggle to use vivid mental imagery for their own problem solving, experience vivid intrusive imagery that interferes with their functioning. This issue needs to be addressed throughout the manuscript.
Finally, please change the sentence describing the comparison between the three alexythymia groups as 'qualitative' and generally make sure that the abstract presents the actual findings of the study, with the direction of associations clear.
Comments on the Quality of English Language
Fine.
Author Response
For research article
|
Response to Reviewer 1 Comments
|
||
|
Summary |
|
|
|
Thank you very much for taking the time to review this manuscript. Please find the detailed responses below and the corresponding revisions/corrections highlighted/in track changes in the re-submitted files. |
||
Comment and Response
Reviewer 1
This manuscript describes a study that brings together three important factors - parental control, mental imagery vividness and alexythymia. It is generally well written and a would make a good scientific contribution. However, it has a flaw in the presentation of the analyses, and it is missing sufficient considerating of a conceptual/psychometric issue. The issue with the analysis is that the tables and text do not show the steps that one would expect from a mediation analysis, and conclusions are made within the text of the Results that intepret and attempt to enhance the meaning of the findings in an overgeneral manner.
This sentence is particularly telling: ". When introducing VVIQ into the regression equation, the predictive relationship between PPC and 274 TAS remained statistically significant (β = 0.33, p < 0.01). This underscores the intricate interplay between PPC, VVIQ, and TAS, thus further supporting the notion that visual mental imagery potentially mediates the relationship between parental psychological control and alexithymia." This is both statistically incorrect, and not appropriate for a Results section. It is not the case that by adding an additional predictor, the maintenance of an existing effect illustrates that the additional predictor is a mediator. I also cannot work out how the following table illustrates mediation. I think it would be easier for the authors to follow the simple Baron and Kenny approach and spell out the steps in the text.
Response: We appreciate the reviewer's feedback. To enhance clarity, we have made adjustments to the results section. Firstly, we omitted the results of the subscales of TAS, as they yielded the same outcomes as the total TAS. Secondly, we adopted the straightforward Baron and Kenny approach, explicating the steps in the text.
The conceptual/psychometric issue is that vivid controllable mental imagery is different from vivid intrusive and uncontrollable mental imagery. I accept that the VVIQ may not measure this, but arguable a new measure needs to be identified. It is often the case that the very same individuals who struggle to use vivid mental imagery for their own problem solving, experience vivid intrusive imagery that interferes with their functioning. This issue needs to be addressed throughout the manuscript.
Response: We appreciate the reviewer's comment. We believe that individuals grappling with challenges in utilizing vivid mental imagery for their problem-solving may also contend with intrusive imagery that hinders their functioning. Our current study specifically focuses on one aspect of the controllable mental imagery—the vividness of visual mental imagery in individuals with alexithymia, who face difficulties in identifying and describing their emotions. In future studies, it would be valuable to explore how vivid, intrusive, and uncontrollable mental imagery influences alexithymia. We have incorporated this aspect into throughout the manuscript.
Finally, please change the sentence describing the comparison between the three alexythymia groups as 'qualitative' and generally make sure that the abstract presents the actual findings of the study, with the direction of associations clear.
Response: We appreciate your guidance in refining the abstract. We have carefully addressed your suggestions and made the necessary revisions. The sentence describing the comparison between the three alexithymia groups has been adjusted accordingly to accurately reflect the nature of the comparison.
Reviewer 2 Report
Comments and Suggestions for Authors
The last part of the Introduction section (last paragraph) should also focus on specifically indicating the objective of the study to complement the indicated study hypotheses.
At the methodological level, the study needs an important development to improve and clarify different basic aspects. First, in the participants section, the inclusion and exclusion criteria considered for participation in the study must be clearly specified. It would also be necessary to indicate the calculation of the number of participants needed to obtain a representative sample, since it is relevant to demonstrate that the sample size is adequate. The total number of participants who were invited to participate and how many were eliminated according to the inclusion and exclusion criteria to form the sample that was finally analyzed should be indicated.
In the instruments section, the use of the Chinese version of the TAS-20 is mentioned, regarding the VVIQ and PPC instruments, specific reference should be made to the Chinese version used or if they have been adapted to carry out this study. On the other hand, it should be specified how the sociodemographic variables, age, sex, or type of student were examined. In this regard, it should be described whether an ad hoc survey or another specific instrument was used.
In section 2 (Materials and Methods) it is essential to include a Procedure subsection. This subsection should include information on approval by the ethics committee and dates of data collection. In this subsection should also specify how this data collection was carried out, indicating whether the evaluation was conducted in person, whether it was conducted in on-line format, control processes used to detect biases, web platforms used (if applicable), etc.
In the Results section it could be relevant to include a complete description of the sociodemographic characteristics of the sample, and to consider variables such as age and sex in the correlation analyses between the psychological variables examined. In the Discussion, it would be appropriate to complement the limitations described with other relevant limitations derived from the procedure description developed in the Materials and Methods section.
Lastly, the citations included in the text should comply with the journal specified regulations, since in some cases there are errors in the citation format used. Also, the "Reference" section should be revised as some of the references included are not complete. The typography of the different sections and subsections titles should be standardized to adapt the text to the journal structural characteristics.
Author Response
For research article
|
Response to Reviewer 2 Comments
|
||
|
Summary |
|
|
|
Thank you very much for taking the time to review this manuscript. Please find the detailed responses below and the corresponding revisions/corrections highlighted/in track changes in the re-submitted files. |
||
Comment and Response
The last part of the Introduction section (last paragraph) should also focus on specifically indicating the objective of the study to complement the indicated study hypotheses.
Response: We express our gratitude to the reviewer for providing valuable feedback. In response to the suggestion, we have incorporated the study's objective into the final paragraph of the introduction section. The added objective is outlined below:
Expanding upon the existing body of research that underscores the connection between parental psychological control, mental imagery, and alexithymia, it is imperative to undertake an empirical investigation into their underlying mechanisms. Consequently, the primary objective of this study is to examine whether college students exhibiting high levels of alexithymia also demonstrate elevated levels of parental psychological control and reduced vividness in visual mental imagery. Additionally, this study puts forth the following hypotheses: that the vividness of mental imagery serves as a mediating factor in the influence of parental psychological control on alexithymia.
At the methodological level, the study needs an important development to improve and clarify different basic aspects. First, in the participants section, the inclusion and exclusion criteria considered for participation in the study must be clearly specified.
Response: We thank the reviewer for the comment. Participants were conveniently recruited from universities, and no other recruitment strategies were employed. University students in four years of undergraduate studies were invited to participate in this study. Students who refused to participate and forms with the wrong answer to the question "somewhat agree" were excluded.
It would also be necessary to indicate the calculation of the number of participants needed to obtain a representative sample, since it is relevant to demonstrate that the sample size is adequate. The total number of participants who were invited to participate and how many were eliminated according to the inclusion and exclusion criteria to form the sample that was finally analyzed should be indicated.
Response: Thank you for your feedback. We have incorporated the sample size calculation into the revised manuscript. For your reference, we provide the citation here:
Sample size calculation
Sample size was calculated using G*Power version 3.1.9.7. Due to different statistical methods applied, we set the parameters for the test with the highest sample size requirement, that is, an F test. Setting a medium effect size (f = 0.25), an α-error probability of 0.05, a power of 0.95, and 3 number of groups, the required sample size was 252 participants.
In the instruments section, the use of the Chinese version of the TAS-20 is mentioned, regarding the VVIQ and PPC instruments, specific reference should be made to the Chinese version used or if they have been adapted to carry out this study.
Response: We appreciate the reviewer's comment, and in response, we have included a specific reference to the Chinese version of TAS-20 in the revised manuscript.
On the other hand, it should be specified how the sociodemographic variables, age, sex, or type of student were examined. In this regard, it should be described whether an ad hoc survey or another specific instrument was used.
Response: Thank you for your valuable comment. We did not use an ad hoc survey or another specific instrument to evaluate the sociodemographic variables. As all the participants were university students, we did not include age information. Instead, we provided details on gender (male or female), grades (Freshman, Sophomore, Junior, Senior), and whether the participant is an only child or not. We have also added this information in the revised manuscript.
In section 2 (Materials and Methods) it is essential to include a Procedure subsection. This subsection should include information on approval by the ethics committee and dates of data collection. In this subsection should also specify how this data collection was carried out, indicating whether the evaluation was conducted in person, whether it was conducted in on-line format, control processes used to detect biases, web platforms used (if applicable), etc.
Response: Thank you for your valuable feedback. We have incorporated the procedure section into our revised manuscript. For your convenience, we provide the relevant citation here:
Procedure
This study obtained approval from the local Ethics Committee of the School of Psychology, South China Normal University (SCNU-PSY-2023-393). Participants were recruited through social media platforms, primarily WeChat. An online survey was designed using Wenjuanxing (https://www.wjx.cn) and shared on social media. The online survey initiated with an informed consent process, during which students were informed about the survey's focus on exploring the impact of emotional difficulties. Subsequently, participants answered demographic questions, followed by self-reported measurements. Data collection occurred from September 2018 to January 2019. Convenient sampling was employed to recruit participants from universities, and no al-ternative recruitment strategies were utilized. University students in four years of undergraduate studies were invited to participate in this study. Students who refused to participate and forms with the wrong answer to the question "somewhat agree" were excluded.
In the Results section it could be relevant to include a complete description of the sociodemographic characteristics of the sample, and to consider variables such as age and sex in the correlation analyses between the psychological variables examined.
Response: We appreciate the reviewer's comment. In response, we have included a detailed description of the sociodemographic characteristics of the sample in the Participant section. Here is the relevant information: "The final analysis included 282 participants, with 52.50% identifying as female. The participant distribution was primarily composed of freshmen (0.7%), sophomores (6.4%), juniors (15.2%), and senior students (77.7%). Additionally, 52.8% of participants were the sole child in their families."
In the Discussion, it would be appropriate to complement the limitations described with other relevant limitations derived from the procedure description developed in the Materials and Methods section.
Response: We thank the reviewer for the comment, and we have addressed this concern by including the limitation related to the exclusion criteria of participants in the revised manuscript. Additionally, we have cited the relevant information:
Finally, other variables associated with psychiatric disorders, the use of medications that affect attention or cognition and diagnosis of any medical illness were not assessed in this study, potentially introducing confounding bias. Future studies should investigate whether other mental illnesses influence the relationship between mental imagery and alexithymia.
Lastly, the citations included in the text should comply with the journal specified regulations, since in some cases there are errors in the citation format used.
Response: We appreciate the reviewer's comment. In response, we have corrected the citation format for all references in the revised manuscript.
Also, the "Reference" section should be revised as some of the references included are not complete. The typography of the different sections and subsections titles should be standardized to adapt the text to the journal structural characteristics.
Response: We appreciate the reviewer's comment. In response, we have corrected the citation format for all references in the revised manuscript.
Reviewer 3 Report
Comments and Suggestions for Authors
I find this study interesting and correctly designed, and the manuscript generally well written. I have some comments that I believe can increase the quality of the paper.
- the Introduction (and/or the Discussion) should provide an explanation of the presumed relationship between parental psychological control and mental imagery. The authors note “we find a limited number of studies directly investigating how the former impacts the latter”. Nevertheless, a plausible account of the psychological dynamics involved in this effect should be presented.
- line 134 “After excluding… the variable values exceeded two standard deviations” – an argument for this decision should be provided.
- it is not clear from the Methods section whether the cut-points on the TAS-20 measure are derived from a validation study of the scale on the Chinese population (as it should in order to use these scores in this research).
- line 208 “Gender, grade and only-child were included as covariates in the model.” – an argument for this should be provided, related to past studies that found differences in the variables included in the mediation model on these demographic variables.
- the paragraph from line 217 should not include means and SDs, as these are reported in Table 1. Similarly, correlations and regression coefficients should not be reported twice.
- the last column in table 1 is not labeled.
- line 306 “. For participants exhibiting elevated levels of alexithymia, visual mental imagery only partially mediated the connection between parental psychological control and difficulty identifying feelings (standardized indirect effect: 0.12, SE=0.08, CIs=0.01-0.34). Conversely, among participants without alexithymia, VVIQ did not mediate any relationships between parental psychological control and the subcomponents of alexithymia” – these analyses are not necessary, as they are performed on homogeneous participants on the dependent variable.
- the explanation of the relation between parental psychological control and alexithymia from line 357 refers to the effect of the latter on the former, which in opposite to the direction of the effect that was included in the model hypothesized by the study. This should be corrected.
- a coherent citing and referring style should be used across the entire manuscript – e.g. line 30 R Michael Bagby, James DA Parker, & Graeme J Taylor, 1994; Parker, Taylor, & Bagby, 2003).
Author Response
For research article
|
Response to Reviewer 3 Comments
|
||
|
Summary |
|
|
|
Thank you very much for taking the time to review this manuscript. Please find the detailed responses below and the corresponding revisions/corrections highlighted/in track changes in the re-submitted files. |
||
Comment and Response
I find this study interesting and correctly designed, and the manuscript generally well written. I have some comments that I believe can increase the quality of the paper.
- the Introduction (and/or the Discussion) should provide an explanation of the presumed relationship between parental psychological control and mental imagery. The authors note “we find a limited number of studies directly investigating how the former impacts the latter”. Nevertheless, a plausible account of the psychological dynamics involved in this effect should be presented.
Response: We express gratitude for the reviewer's comment, and as a thoughtful response, we have integrated an additional reference delving into the impact of adolescents' perceived negative evaluation of parenting on their visuo-spatial attention and mental rotation abilities. The referenced study posits that the negative evaluation of parenting attitudes by adolescents might instigate a noticeable decline in cognitive abilities, particularly in visuo-spatial attention and mental rotation, preceding the formal diagnosis of related mental disorders. This proposed mechanism formed the foundational basis of our study.
- line 134 “After excluding… the variable values exceeded two standard deviations” – an argument for this decision should be provided.
Response: Outliers (n = 23) were defined as those for whom the discrepancy between self-reported and measured variables exceeded two standard deviations from the mean discrepancy. The identification and removal of outliers served to remove their potentially undue influence during correction generation. Results were the same with and without outliers.
- it is not clear from the Methods section whether the cut-points on the TAS-20 measure are derived from a validation study of the scale on the Chinese population (as it should in order to use these scores in this research).
Response: Thank you for highlighting this concern. The cut-off of 61 was recommended by the scale's author, and numerous studies, including those involving Chinese participants, have consistently utilized this threshold. Aligning with established practices, we adopted 61 as the cut-off value in our study. To enhance transparency, we will explicitly convey these details in the Methods section, supported by references, ensuring a clear and well-documented explanation.
- line 208 “Gender, grade and only-child were included as covariates in the model.” – an argument for this should be provided, related to past studies that found differences in the variables included in the mediation model on these demographic variables.
Response: We appreciate the reviewer's comment. In response, we have included relevant references to support and demonstrate the impact of gender, grade, and only-child status on alexithymia in our revised manuscript.
- the paragraph from line 217 should not include means and SDs, as these are reported in Table 1. Similarly, correlations and regression coefficients should not be reported twice.
Response: We appreciate your observation, and we have revised the paragraph to exclude means and SDs, and avoid reporting correlations and regression coefficients twice.
- the last column in table 1 is not labeled.
Response: Thank you for bringing this to our attention. We have adjusted the format of Table 1 to include a label for the last column.
- line 306 “. For participants exhibiting elevated levels of alexithymia, visual mental imagery only partially mediated the connection between parental psychological control and difficulty identifying feelings (standardized indirect effect: 0.12, SE=0.08, CIs=0.01-0.34). Conversely, among participants without alexithymia, VVIQ did not mediate any relationships between parental psychological control and the subcomponents of alexithymia” – these analyses are not necessary, as they are performed on homogeneous participants on the dependent variable.
Response: Thank you for providing feedback on the analyses presented in line 306. After careful consideration, we agree that these analyses do not significantly contribute to the understanding of our research question. Consequently, we have excluded them from our study.
- the explanation of the relation between parental psychological control and alexithymia from line 357 refers to the effect of the latter on the former, which in opposite to the direction of the effect that was included in the model hypothesized by the study. This should be corrected.
Response: Thank you for bringing this to our attention. We appreciate your thorough review. We acknowledge the inconsistency in the explanation of the relation between parental psychological control and alexithymia from line 357, where the described effect contradicts the direction specified in the study's hypothesized model. In response to your feedback, we have revised the sentence that reads, "A plausible explanation is that the alexithymia trait, defined by a cognitive style that avoids introspection, may impact emotion generation and have the potential to hinder the construction of internal representations of emotional stimuli and actions." The updated version now reads: "A plausible explanation is that the compromised construction of internal representations of emotional stimuli in alexithymia may serve as a mechanism explaining their challenges in emotional processing." We hope this clarification addresses the issue.
- a coherent citing and referring style should be used across the entire manuscript – e.g. line 30 R Michael Bagby, James DA Parker, & Graeme J Taylor, 1994; Parker, Taylor, & Bagby, 2003).
Response: Thank you for your valuable feedback. We have implemented the necessary changes in the manuscript. The author adopted the Toronto Alexithymia Scale in 2003, prompting us to replace the outdated 1994 version with the updated reference. The suggested reference format has been seamlessly integrated to enhance clarity and maintain consistency throughout the manuscript. Additionally, we have removed references that were deemed too outdated, contributing to a more streamlined and focused presentation.
Round 2
Reviewer 2 Report
Comments and Suggestions for Authors
The author has made careful revisions and responses, and the quality of the article has improved.
Reviewer 3 Report
Comments and Suggestions for Authors
The authors addressed my previous comments.